# Comparison of Hospital-Based and Home-Based Obstructive Sleep Apnoea Severity Measurements with a Single-Lead Electrocardiogram Patch

**DOI:** 10.3390/s21238097

**Published:** 2021-12-03

**Authors:** Wen-Te Liu, Shang-Yang Lin, Cheng-Yu Tsai, Yi-Shin Liu, Wen-Hua Hsu, Arnab Majumdar, Chia-Mo Lin, Kang-Yun Lee, Dean Wu, Yi-Chun Kuan, Hsin-Chien Lee, Cheng-Jung Wu, Wun-Hao Cheng, Ying-Shuo Hsu

**Affiliations:** 1School of Respiratory Therapy, College of Medicine, Taipei Medical University, Taipei 110301, Taiwan; lion5835@gmail.com (W.-T.L.); young19820822@gmail.com (S.-Y.L.); m141109002@tmu.edu.tw (Y.-S.L.); 2Sleep Center, Shuang Ho Hospital, Taipei Medical University, New Taipei City 235041, Taiwan; tingyu02139@gmail.com (D.W.); yckuang2@gmail.com (Y.-C.K.); 3Division of Pulmonary Medicine, Department of Internal Medicine, Shuang Ho Hospital, Taipei Medical University, New Taipei City 235041, Taiwan; leekangyun@tmu.edu.tw; 4Department of Civil and Environmental Engineering, Imperial College London, London SW7 2AZ, UK; ct619@ic.ac.uk (C.-Y.T.); a.majumdar@imperial.ac.uk (A.M.); 5Master Program, Thoracic Medicine School of Respiratory Therapy, College of Medicine, Taipei Medical University, Taipei 110301, Taiwan; b117105061@tmu.edu.tw; 6Sleep Center, Shin Kong Wu Ho-Su Memorial Hospital, Taipei 111045, Taiwan; aminus64@gmail.com; 7Division of Pulmonary Medicine, Department of Internal Medicine, School of Medicine, College of Medicine, Taipei Medical University, Taipei 110301, Taiwan; 8Department of Neurology, Shuang Ho Hospital, Taipei Medical University, New Taipei City 235041, Taiwan; 9Department of Neurology, School of Medicine, College of Medicine, Taipei Medical University, Taipei 110301, Taiwan; 10Taipei Neuroscience Institute, Taipei Medical University, Taipei 110301, Taiwan; 11Dementia Center, Shuang Ho Hospital, Taipei Medical University, New Taipei City 235041, Taiwan; 12Department of Psychiatry, Taipei Medical University Hospital, Taipei 110301, Taiwan; ellalee@tmu.edu.tw; 13Department of Otolaryngology, Shuang Ho Hospital, Taipei Medical University, New Taipei City 235041, Taiwan; 14667@s.tmu.edu.tw; 14Biomedical Science and Engineering, National Yang Ming Chiao Tung University, Hsinchu 300093, Taiwan; 15Graduate Institute of Medical Sciences, College of Medicine, Taipei Medical University, Taipei 110301, Taiwan; d119106011@tmu.edu.tw; 16Department of Otolaryngology, Shin Kong Wu Ho-Su Memorial Hospital, Taipei 111045, Taiwan; 17School of Medicine, Fu Jen Catholic University, Taipei 242062, Taiwan; 18Institute of Brain Science, National Yang Ming Chiao Tung University, Taipei 112304, Taiwan

**Keywords:** apnoea–hypopnea index (AHI), cyclic variation of heart rate index (CVHRI), obstructive sleep apnoea (OSA), polysomnography (PSG)

## Abstract

Obstructive sleep apnoea (OSA) is a global health concern, and polysomnography (PSG) is the gold standard for assessing OSA severity. However, the sleep parameters of home-based and in-laboratory PSG vary because of environmental factors, and the magnitude of these discrepancies remains unclear. We enrolled 125 Taiwanese patients who underwent PSG while wearing a single-lead electrocardiogram patch (RootiRx). After the PSG, all participants were instructed to continue wearing the RootiRx over three subsequent nights. Scores on OSA indices—namely, the apnoea–hypopnea index, chest effort index (CEI), cyclic variation of heart rate index (CVHRI), and combined CVHRI and CEI (Rx index), were determined. The patients were divided into three groups based on PSG-determined OSA severity. The variables (various severity groups and environmental measurements) were subjected to mean comparisons, and their correlations were examined by Pearson’s correlation coefficient. The hospital-based CVHRI, CEI, and Rx index differed significantly among the severity groups. All three groups exhibited a significantly lower percentage of supine sleep time in the home-based assessment, compared with the hospital-based assessment. The percentage of supine sleep time (∆Supine%) exhibited a significant but weak to moderate positive correlation with each of the OSA indices. A significant but weak-to-moderate correlation between the ∆Supine% and ∆Rx index was still observed among the patients with high sleep efficiency (≥80%), who could reduce the effect of short sleep duration, leading to underestimation of the patients’ OSA severity. The high supine percentage of sleep may cause OSA indices’ overestimation in the hospital-based examination. Sleep recording at home with patch-type wearable devices may aid in accurate OSA diagnosis.

## 1. Introduction

Obstructive sleep apnoea (OSA) is a major health concern in modern society. A systematic review published in 2017 reported that OSA prevalence ranges between 9% and 38% in the general population [1]. Moreover, OSA has been demonstrated to be associated with several comorbidities, including metabolic syndrome, cardiovascular diseases, and neurodegenerative diseases [2,3]. Polysomnography (PSG) is the gold standard for determining a patient’s apnoea–hypopnea index (AHI), which is calculated by the number of respiratory events (apnoea and hypopnea) per hour of sleep and used to classify OSA severity. However, PSG is complicated and inconvenient to implement. Patients typically undergo PSG with multiple leads on their bodies at a hospital sleep centre. The discomfort involved in the PSG itself can cause sleep disturbance. Moreover, relevant studies have indicated that the sleep parameters obtained using PSG could be underestimated or overestimated because of environmental factors or the first-night effect [4].

Therefore, to assess OSA severity in contexts where the results are less likely to be affected by environmental factors, the American Academy of Sleep Medicine has classified unattended home monitoring devices into three types [5]. The most common home sleep apnoea testing devices available are type-3 devices that include parameters for evaluating the respiratory status, cardiac function, and pulse oxygen saturation (SpO_2_). Type-4 devices consider one or two of these parameters. Studies have applied wearable devices operated using various technologies, such as actigraphy, finger-based pulse oximetry, and single-channel electrocardiography, for the home-based recording of sleep parameters [6,7,8]. More recently, a biosensor that integrates an electrocardiography module and a three-axis accelerometer was developed, demonstrating favourable reliability and accuracy in evaluating OSA severity [9]. Although various wearable devices have been used for OSA severity assessment, uncertainties remain regarding the differences between hospital-based PSG parameters and home-based sleep variables.

Several investigations have compared home sleep apnoea testing results and hospital-based sleep parameters. One study suggested that home-based AHI values are underestimated relative to hospital-based AHI values [10]. This disparity may be attributed to the lack of sleep stage measurement at home, which leads to the overestimation of total sleep time. Another study reported night-to-night variability in the results of hospital-based PSG, with relatively weak correlations between test–retest AHI values [11]. Hospital-based AHI measurements might not accurately represent sleep status because they are easily affected by the different percentages of time spent in various sleeping positions depending on the scenario [12]. Consequently, uncertainty remains regarding the association between home-based and hospital-based measurements of OSA severity. Moreover, to the best of our knowledge, no studies have undertaken the acquisition of long-term sleep parameters or conducted in-depth evaluations based on variations in sleeping positions.

To determine long-term home sleep parameters and prevent sleep disturbance caused by cumbersome instruments, the cyclic variation of heart rate index (CVHRI) can be used as a potential surrogate for screening OSA severity. This index is calculated according to the specific heart rate alternation in progressive bradycardia when an apnoea event occurs and is followed by abrupt tachycardia on breathing resumption [13]. Several relevant studies have been performed to improve the algorithm and validate the associations between CVHRI and AHI [14,15,16]. CVHRI can also be directly determined by analysing single-lead electrocardiogram (ECG) signals [14]. The chest effort index (CEI), determined through assessment of a patient’s chest wall motion, is a potential surrogate marker for OSA risk. Chest wall motion is directly affected by respiratory events during sleep; that is, chest wall movement is reduced when an apnoeic event occurs. Studies have asserted that sleep-disordered breathing events are characterised by chest wall distortion and paradoxical chest wall movement caused by the respiratory effort against airway obstruction [17,18]. Hence, the CVHRI and CEI are potential alternative indices for the non-invasive observation of sleep parameters over multiple days.

This study examined the use of patch-type wearable devices for determining patients’ OSA indices and sleep positions in a hospital (during PSG examination) and during three nights at home following hospital discharge. Next, correlation analysis and statistical examinations were conducted to investigate the differences and relationships among the derived data. The primary objective of this study was to compare the data on sleep parameters obtained in overnight PSGs conducted at the hospital and over several days at home by using a single-lead ECG patch with a three-axis accelerometer (RootiRx), and the results are expected to enhance our understanding of how sleep positions and the environment affect OSA severity. Specifically, the observed outcomes can support the hypothesis that having a high percentage of sleep time spent in supine sleeping is correlated with severe OSA. These findings also highlighted the possible use of patch-type wearable devices to personalise OSA treatment options (e.g., patients who tend to experience positional OSA can consider positional therapy as a treatment option rather than other low-adherence options such as continuous positive airway pressure). Furthermore, we investigated the changes in sleeping position in various sleep environments to determine the correlations between the percentage of sleep time spent in the supine position and OSA severity for both in-laboratory PSG and RootiRx assessments. The derived results suggested that hospital PSG devices can aggravate OSA severity because they limit the extent to which patients can change their sleep positions.

## 2. Related Research

Several studies have discussed the association between sleep position and demonstrated OSA severity. For example, relevant research analysed hospital-based PSG data and indicated that AHI was significantly associated with sleeping in the supine position [19]. A study examined the sleep parameters of 571 participants and reported that OSA was influenced by the time spent sleeping in the supine position for more than 50% of the participants [20]. However, the possibility that PSG leads to a high percentage of sleep time spent in the supine position and directly causes OSA severity overestimation requires further clarification. In other words, based on the aforementioned prior findings, OSA severity, which is considerably affected by the sleep position, may be aggravated by the sleep position restrictions imposed by PSG devices because of their complexity. Hence, comparing the sleep parameters measured at home and in the hospital may help to determine the effect of PSGs on OSA severity.

In recent years, wearable technology has developed rapidly, mainly through the application of existing technologies for developing miniaturised and portable devices. In sleep research, various contact biosensors are used to measure physiological signals (e.g., ECG or SpO_2_ variation) to determine the sleep profile. For example, a study established models using multiple machine learning techniques to examine single-lead ECG signals, thereby allowing for OSA diagnoses to be performed with high accuracy [21]. Another study employed oximetry to perform long-term sleep profile tracking [22]. The use of these novel wearable devices complemented OSA diagnosis with considerably less interference. Nevertheless, few studies have compared hospital- and home-based sleep parameters by using validated devices that interfere less with sleep.

To address the limitations of related studies, the present study used a validated patch-type wearable device that interferes less with sleep to obtain OSA indices in the hospital (during PSG examinations) and at home (three nights following hospital discharge). By examining the correlations between the sleep time in the supine position and the OSA index values on the basis of the hospital- and home-based sleep parameters, this study may partially reveal that PSG causes a high percentage of sleep time to be spent in the supine position and is also related to OSA severity overestimation. Furthermore, this study may further clarify the feasibility of using patch-type wearable devices to aid OSA treatment suggestions, particularly for patients with positional OSA.

## 3. Materials and Methods

### 3.1. Study Population

We recruited patients with reported snoring or with suspected sleep-disordered breathing who were referred to the sleep centres of SKH and SHH between February 2018 and January 2019. The inclusion criteria were as follows: patients (1) aged between 18 and 80 years who (2) were not pregnant, (3) did not have a diagnosis of other cardiovascular diseases, and (4) had a total PSG recording time of >6 h. To reduce the possibility of OSA severity overestimation caused by short sleep time in the hospital setting, a large proportion of patients (103 of 125) with high sleep efficiency were recruited from the sleep centres to form 2 subgroups (≥80% and ≥90%). Sleep efficiency is commonly defined as the ratio of a patient’s total sleep time to total time in bed and is generally utilised as an index for the evaluation of sleep quality. The patients underwent PSG while wearing a wireless single-lead ECG monitoring patch (RootiRx, Rooti Labs, Taipei, Taiwan). After PSG was completed, the patients were instructed to continue wearing the provided patch over 3 subsequent nights to collect relevant sleep parameters under home sleep conditions. All the values for the hospital- and home-derived sleep parameters were used for further analysis and comparison.

### 3.2. PSG Results

PSG is a systematic process through which (1) physiological parameters are collected during sleep and (2) the underlying causes of sleep disorders are assessed on the basis of various physiological signals. Notably, PSG is considered a standard method for diagnosing sleep-related breathing disorders, including OSA, central sleep apnoea, and sleep-related hypoventilation or hypoxia [23]. We obtained PSG recordings by using the Compumedics Grael PSG system (SKH) or the ResMed Embla N7000 and Embla MPR systems (SHH). We scored the sleep stages and respiratory events according to the updated standard diagnostic criteria and scoring guidelines of the American Academy of Sleep Medicine [24,25]. Licensed PSG technicians scored the results at both sleep centres, and these scores were confirmed by at least 2 other technicians to ensure accuracy. We determined the AHI value of each patient to classify them into the following 3 groups: no-to-mild (AHI < 15 events/h), moderate (15 ≤ AHI < 30 events/h), and severe (AHI ≥ 30 events/h) OSA.

### 3.3. Home Sleep Recording

We obtained the home sleep parameters through observation with RootiRx. The technical details of this device and the definition of the obtained sleep parameters, including CVHRI, CEI, and combined CVHRI and CEI (Rx index), were documented in our previous study [26]. In the current study, CVHRI, CEI, and Rx index were determined first at the sleep centres through PSG and subsequently at home for 3 consecutive nights. The triaxial accelerometer in the device assessed the percentage of sleep time spent in different positions. All of the derived data were then separated into hospital and home data groups for comparison.

### 3.4. Statistical Analysis

All statistical analyses, the framework of which is presented in Figure 1, were conducted using SPSS software (IBM SPSS Inc., Chicago, IL, USA). First, we conducted the Shapiro–Wilk test to examine the normality of the continuous variables. The baseline characteristics of the patients in the OSA groups were compared, using one-way analysis of variance (normally distributed data) or the Kruskal–Wallis test (nonnormally distributed data) for the continuous variables and the chi-squared test for the categorical variables. Subsequently, we performed Student’s *t*-test (normally distributed data) or the Mann–Whitney U test (nonnormally distributed data) to compare the sleep parameters and positions obtained at the sleep centres and at home. The correlations between the variations in the percentage of sleep time spent in a supine position (∆Supine%), and CVHRI (∆CVHRI), CEI (∆CEI), and Rx index (∆Rx index) were investigated through the Spearman rank correlation test. All tests were two-tailed, and differences were considered significant at *p* < 0.05.

## 4. Results

### 4.1. Sample Characteristics

A total of 125 patients were included. Table 1 presents the baseline characteristics of the patients according to OSA severity (segmented based on values of hospital-based AHI). In the sample, 33, 31, and 61 patients were classified as having no-to-mild OSA, moderate OSA, and severe OSA, respectively. No significant differences in age or sex were noted among the three groups. Regarding the participants’ body profiles, a significantly higher body mass index and higher neck circumference were observed in the moderate and severe OSA groups. Regarding the hypoxemia-related indicators, mean SpO_2_, minimum SpO_2_, and oxygen desaturation index (≥3%) were significantly lower in the severe OSA group than in the moderate OSA and no-to-mild OSA groups. Overall, the patients in the severe OSA group had obesity (higher body mass index and neck circumference values) and poorer sleep oxygenation (lower values for mean and lowest SpO_2_; higher values for oxygen desaturation index and AHI) relative to the other groups.

### 4.2. Hospital- and Home-Based Sleep Parameters

Table 2 and Figure 2 present the variations in the hospital- and home-based sleep parameters. Notably, the severe OSA group exhibited significantly higher means in the hospital-based measurement than in the home-based measurement (CVHRI, hospital: 33.8 ± 21.1 events/h, home: 20.4 ± 18.2 events/h; CEI, hospital: 18.2 ± 12.2 events/h, home: 13.1 ± 9.9 events/h; Rx index, hospital: 40.9 ± 21.1 events/h, home: 27.1 ± 18.4 events/h). In all groups, the percentage of supine sleep time in the hospital setting (range: 72.5%–74.1%) was significantly higher than that in the home setting (range: 48.9%–58.0%). Moreover, OSA severity, as classified by AHI in the hospital setting, decreased to a milder grade at home in 54.4% and 41.6% of participants in classification by CVHRI and CEI, respectively. Collectively, the patients in the severe group exhibited significantly higher values for hospital-based OSA indices (CVHRI, CEI, and Rx index), compared with home-based values. All patients exhibited a significantly lower percentage of sleep time spent in the supine position at home, compared with the values measured in the hospital.

### 4.3. Sleeping Position and Sleep-Related Indices

Figure 3 illustrates the correlations between the ∆Supine% and sleep-related parameters (∆CVHRI, ∆CEI, and ∆Rx index). The ∆Supine% exhibited significant but weak-to-moderate positive correlations with the ∆CVHRI (ρ = 0.27, 95% confidence interval [CI]: 0.10 to 0.43, *p* < 0.01), ∆CEI (ρ = 0.29, 95% CI: 0.12 to 0.44, *p* < 0.01), and ∆Rx index (ρ = 0.33, 95% CI: 0.16 to 0.48, *p* < 0.01). In brief, these statistical outcomes between the sleep percentages and OSA indices suggested that a greater variation in the supine position between hospital- and home-based results increases the variation in the values of OSA indices. Specifically, these findings suggested that the hospital-based OSA indices were overestimated because of the higher percentage of sleep time spent in the supine position.

### 4.4. Variations in Sleeping Position and Sleep-Related Indices in Patients with High Sleep Efficiency

Table 3 presents the alterations in sleep parameters determined in the hospital and home settings in patients with high sleep efficiency (>80% in the hospital-based PSG). Notably, 103 patients had high sleep efficiency, including 24, 24, and 55 patients in the no-to-mild, moderate, and severe OSA groups. Significant differences in CVHRI, CEI, Rx index, and percentage of supine sleep time between the hospital- and home-based measurements were noted in the severe OSA group. In the moderate OSA group, the home-based Rx index and percentage of supine sleep time were significantly lower than those measured in the sleep centres. Moreover, in patients with a sleep efficiency of > 90% (*n* = 46), the home-based measurements of the means of the Rx index and percentage of supine sleep time were significantly lower than the corresponding hospital-based measurements (Figure 4). Collectively, the current observations indicated that patients who exhibited high sleep efficiency and who tended to be less affected by environmental factors still exhibited higher values for hospital-based parameters (including OSA indices and supine sleep time) relative to home-based parameters.

### 4.5. Correlations between Sleep-Related Indices and Sleeping Position in Patients with High Sleep Efficiency

Figure 5 presents the correlation between the ∆Supine% and ∆Rx index in patients with high sleep efficiency. In patients with a sleep efficiency of ≥80%, the ∆Rx index was significantly and moderately correlated with ∆Supine% (ρ = 0.35, 95% CI: 0.17 to 0.51, *p* < 0.01). In patients with a sleep efficiency of ≥90%, the correlation was weaker but still significant (ρ = 0.29, 95% CI: 0.05 to 0.50, *p* < 0.01). Collectively, given that environmental factors can affect OSA severity, patients with high sleep efficiency still exhibited weak-to-moderate relationships between the supine sleep time and OSA indices. These outcomes suggest that the hospital-based OSA indices were overestimated because of the high percentage of sleep time spent in the supine position.

## 5. Discussion

This study compared OSA index values and the percentage of supine sleep time using the RootiRx device in both hospital and home settings. As patch-type wearable devices cause less interference to individuals’ sleep than PSG measurements, lower values for parameters such as CVHRI, CEI, Rx index, and percentage of supine sleep time were noted in the home setting (Table 2 and Figure 2). Moreover, ∆Supine% exhibited a significant but weak-to-moderate positive correlation with each of the OSA indices (Figure 3). The participants with a sleep efficiency of ≥90% in the hospital setting also exhibited higher ∆Rx index and ∆Supine% values in the hospital setting (Figure 4). Furthermore, even among patients with sleep efficiencies of ≥80% and ≥ 90% in the hospital setting, ∆Rx index values still exhibited a significant but weak-to-moderate positive correlation with ∆Supine% (Figure 5). This indicates that patch-type wearable devices such as RootiRx may cause less interference to patients’ sleeping positions than PSG, even when the patients exhibit high sleep efficiency in hospital settings.

Regarding the relationship between sleeping position and OSA indices in hospital- and home-based data, similar outcomes to this study have been previously documented. For example, Kukwa et al. analysed female participants’ sleep parameters and determined that the ∆Supine% recorded by a home sleep apnoea testing was significantly higher than that recorded using PSG [27]. Similarly, Bignold et al. developed a supine avoidance device and reported that inhibition of a patient’s supine sleep decreased their AHI [28]. Compared with these studies, our investigation more straightforwardly examined the relationships between OSA indices and sleep positions in distinct hospital and home environments, culminating in the discovery of a positive correlation between OSA indices and ∆Supine%. This significant positive correlation may contribute to a greater understanding of the effect of sleep position on OSA severity.

Regarding the comparison of home-based and hospital-based sleep parameters, the OSA index values recorded in the hospital were generally higher than those recorded at home, especially among the patients with severe OSA. This implies that in-laboratory examination may overestimate OSA severity, potentially due in part to environmental factors such as the equipment, testing room, and bed, which may hinder changes in sleeping position.

Discrepancies between OSA severity measurements obtained in hospitals and at home have been documented. A study conducted in 1996 centred on the effects of PSG on sleeping position, suggesting that PSG may influence the diagnosis of positional OSA [29]. In that study, 12 patients with positional OSA who had undergone standard PSG returned for 2 additional nights of study without the attachment of PSG leads. The mean percentage of supine sleep time (56%) was greater during the PSG night than during the non-PSG nights. A large-scale retrospective study (2019) on positional OSA treatment with the sleep position trainer, a vibrating device, reported that the PSG apparatus caused an increase in the percentage of supine sleep time and may increase the measured OSA severity [30]. The median AHI decreased from 13.3/h to 10.3/h (*p* < 0.001), and 33% of the patients exhibited a change in OSA severity (AHI obtained in hospital settings vs. adjusted AHI obtained at home). These outcomes support our findings that PSG measurements may affect the sleeping positions and increase the percentage of sleep time in the supine position. Therefore, the effects of PSG equipment on sleeping positions may lead to higher AHI values, leading to the overestimation of OSA severity.

The significant but weak-to-moderate correlations between the ∆Supine% and OSA indices indicate that the increase in ∆Supine% and the corresponding increase in OSA severity might be a general pattern rather than limited to specific patient groups. This finding has clinical relevance because the potentially significant overestimation of OSA severity due to the influence of the PSG apparatus on sleeping position is likely to affect treatment strategies. For instance, a high continuous positive airway pressure setting may be employed in consideration of a patient’s overestimated AHI values, but this may result in the patient’s discontinuation of the therapy due to discomfort. A high continuous positive airway pressure setting may even cause central sleep apnoea in some patients due to elevated carbon dioxide excretion [31]. Furthermore, the individuals who demonstrated a high ∆Supine% in the hospital but low values at home may have positional OSA, and they may therefore be able to consider positional therapy as a treatment option [32].

Some may argue that OSA severity, as determined through PSG, may be affected by other environmental factors. To address this concern, we analysed the sleep parameters in participants with high sleep efficiency, in whom the possibility of OSA severity overestimation owing to sleep stage can be largely excluded. Therefore, patients had long sleep times in the hospital setting, the AHI values obtained from PSG and the OSA index values obtained from the RootiRx were not categorised according to short sleep time. In other words, in patients with high sleep efficiency, the overestimation of OSA severity is more likely to be attributable to sleeping position than to alterations in total sleep time. Moreover, the ∆Rx index was significantly but weakly to moderately correlated with the ∆Supine% in the high sleep efficiency groups. These results suggest that the overestimation of OSA severity in hospitals may be mainly due to patients’ sleeping positions. Therefore, the home-based OSA index values likely represent the participants’ actual OSA severity because they were not restricted by cumbersome PSG devices and could freely alter their sleeping position.

This study has some limitations. First, the RootiRx dataset lacked sleep staging measurements obtained through electroencephalography. The RootiRx device determines the sleep stage of the wearer by using a validated algorithm, such as a fast Fourier transform and neural networks [33,34,35]. Although the accuracy of the predicted sleep stage and estimated total sleep time was approximately 85% to 90%, the arousal response or the precise percentages of rapid eye movement sleep and nonrapid eye movement sleep could not be obtained. The associations between the first-night effect, REM latency, and duration should be further explored [36]. Second, during the RootiRx recording, the effects of environmental factors in the hospital and home sleep environments could not be controlled. Environmental factors include radiant temperature, air temperature, relative humidity, carbon dioxide concentration, illumination, and equivalent noise level [37]. Regarding the inter-and intravariability of the measurements taken using patch-type wearable devices, the sleep parameters were automatically calculated using a validated algorithm to address the concern of intervariability. However, this study did not account for the intravariability of the patch-type devices, which could have been addressed through the repeated collection of the sleep parameters in the home environment. The fluctuating sleep profiles exhibited by each participant each night limited our ability to account for intravariability. Therefore, further studies that incorporate study protocols for repeated measurements are necessary to determine the effect of the intravariability of patch-type wearable devices. Moreover, ECG signals are the mechanism of the RootiRx device. Thus, the CVHRI index could have been affected by abnormal heart rhythms, such as atrial fibrillation and ventricular tachycardia (with or without pacemaker implantation), and arrhythmia caused by any other type of cardiovascular condition. Hence, in patients with related heart diseases, the Rx index and CVHRI might not be accurate measures of OSA. In such patients, CEI may be a more suitable parameter for diagnosing OSA.

Another limitation is the strength of the determined correlation. Specifically, only weak or weak-to-moderate positive correlations were observed between the OSA indices and ∆Supine% in various sleep environments. These outcomes provide only limited evidence suggesting that a high variation between the percentage of sleep time spent in the supine position in the hospital and at home can influence AHI values and indirectly lead to a misestimation of OSA severity. To robustly assess the effect of the sleep position on OSA severity, more data and results indicating a stronger correlation are required.

## 6. Conclusions

This study compared hospital- and home-based sleep parameters by utilising the RootiRx, which is a wearable device that uses a single-lead ECG patch. The current results indicate that patients may spend a higher percentage of sleep time in the supine position in hospital settings, which can partially result in the overestimation of OSA severity. Furthermore, the results suggest that the implementation of home-based sleep recording involving the use of patch-type wearable devices leads to less interference and restrictions with respect to the sleep position; therefore, this tool can complement hospital-based examinations and enhance the accuracy of OSA diagnoses.

## Figures and Tables

**Figure 1 sensors-21-08097-f001:**
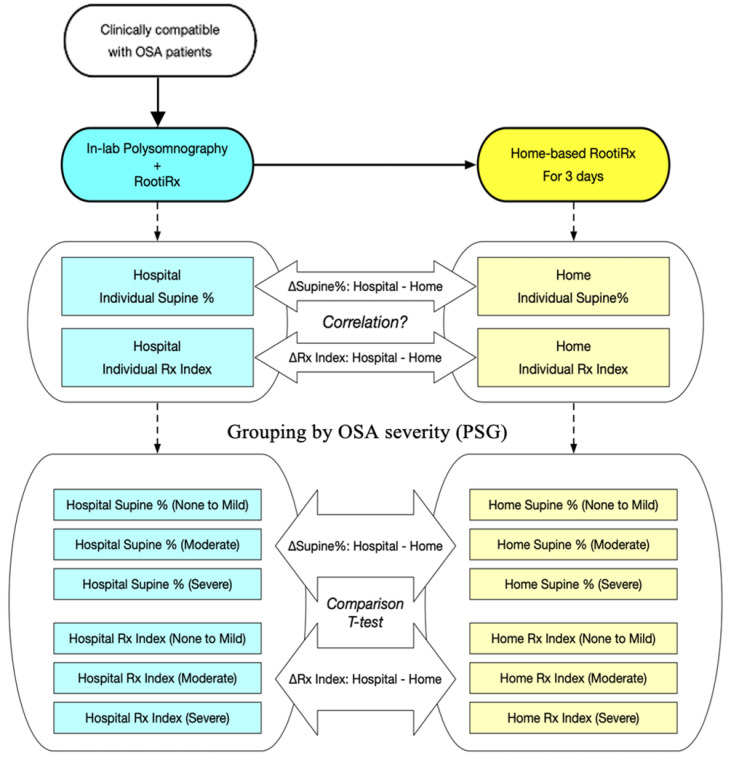
Study framework. Abbreviations: OSA, obstructive sleep apnoea; PSG, polysomnography; Rx index, combination of the cyclic variation of heart rate index and the chest effort index; ∆Supine%, the variations in the supine percentage; ∆Rx index, the variations in Rx index.

**Figure 2 sensors-21-08097-f002:**
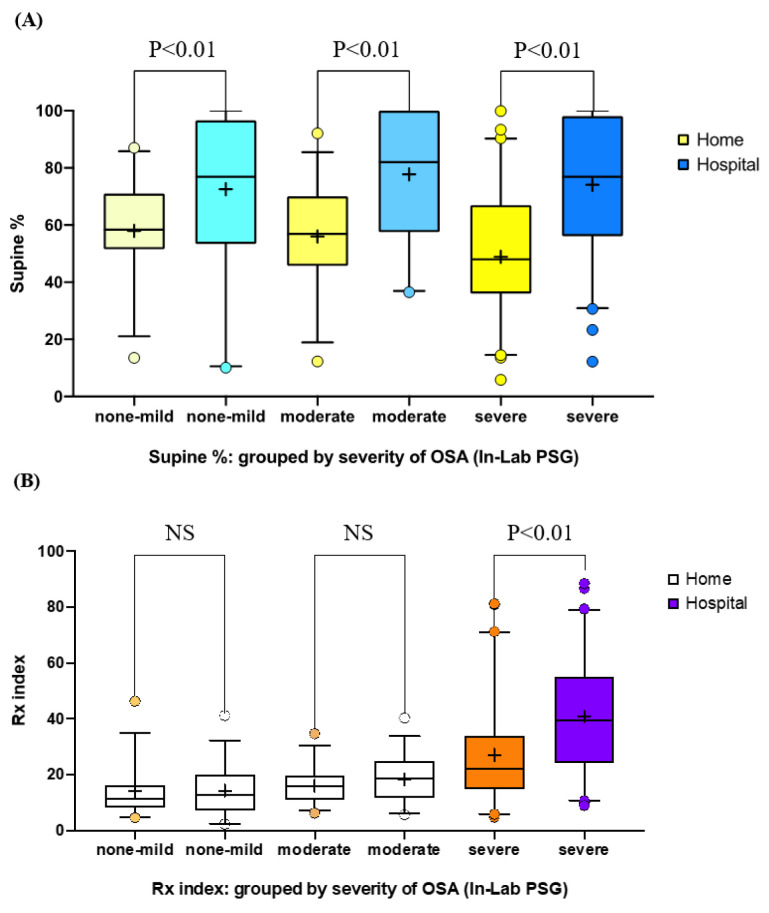
Comparison of the variation in the percentage of supine sleep time and Rx index values obtained in hospital and home settings: (**A**) percentage of supine sleep time in the various OSA severity groups; (**B**) Rx index in the various OSA severity groups. Abbreviations: Rx index, combination of the cyclic variation of heart rate index and the chest effort index; OSA, obstructive sleep apnea; PSG, polysomnography; ns, nonsignificant.

**Figure 3 sensors-21-08097-f003:**
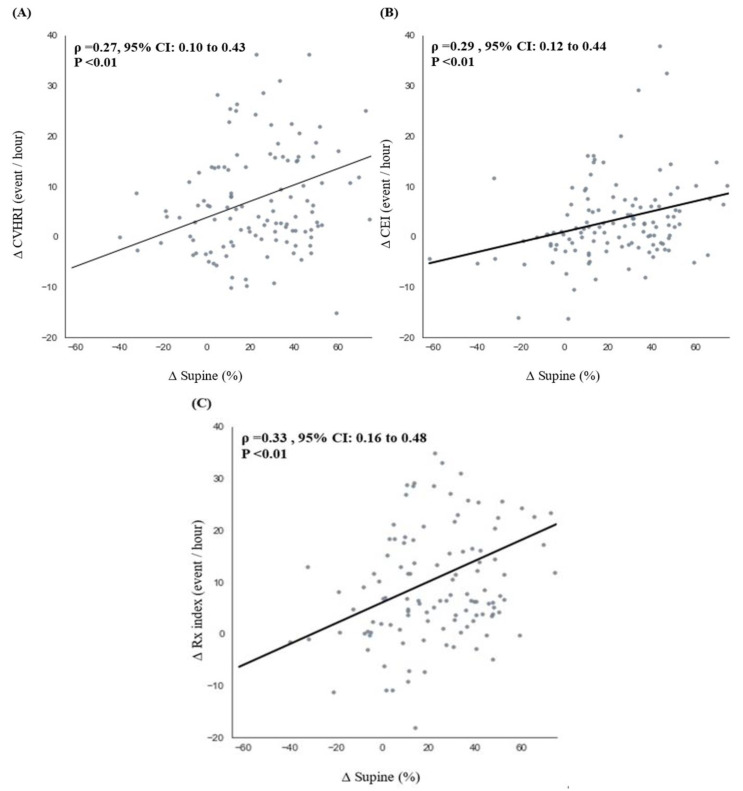
Correlations between the variations in the hospital- and home-based measurements of the percentage of supine sleep time and the RootiRx parameters. Correlation between CVHRI (**A**), CEI (**B**), Rx index (**C**), and variations in the percentage of supine sleep time. Abbreviations: CVHRI, cyclic variation of heart rate index; CEI: chest effort index; Rx index, combination of CVHRI and CEI.

**Figure 4 sensors-21-08097-f004:**
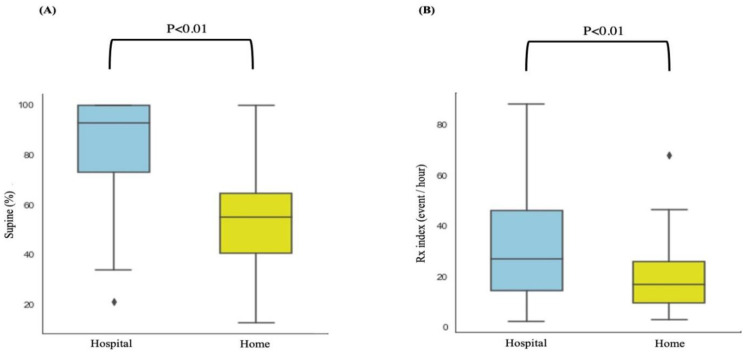
Variation of the percentage of supine sleep time and the Rx Index values determined in hospital and home settings in the high-sleep-efficiency group (≥90%; *n* = 46): (**A**) variation in the percentage of supine sleep in various sleep environments; (**B**) variation in Rx index values in various sleep environments. Abbreviations: Rx index, combination of the cyclic variation of heart rate index and the chest effort index.

**Figure 5 sensors-21-08097-f005:**
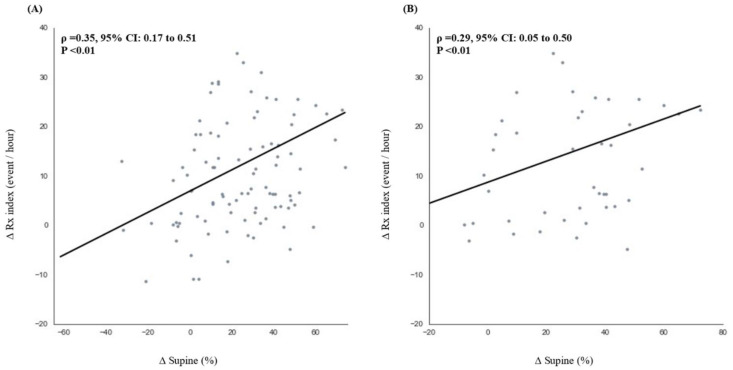
Correlations between the variations between hospital- and home-based measurements of the percentage of supine sleep time and the RootiRx parameters in the high-sleep-efficiency groups: (**A**) correlation between the variations in the percentage of supine sleep time and Rx index values in patients with a sleep efficiency of ≥80% (*n* = 103); (**B**) correlation between the variations in the percentage of supine sleep time and Rx index values in patients with a sleep efficiency of ≥90% (*n* = 46). Abbreviations: Rx index, combination of the cyclic variation of heart rate index and the chest effort index.

**Table 1 sensors-21-08097-t001:** Demographic characteristics of participants grouped according to OSA severity and assessed through hospital-based PSG.

Categorical Variables	No-to-Mild Group (*n* = 33)	Moderate Group (*n* = 31)	Severe Group (*n* = 61)	*p* Value
Age (year)	42.0 ± 11.0	45.3 ± 12.4	45.2 ± 12.8	0.44 ^a^
Body mass index (kg/m^2^)	24.4 ± 2.9	25.5 ± 3.7	28.7 ± 4.5	<0.01 ^b^
Sex (male/female)	22/11	23/8	51/10	0.16 ^c^
Neck circumference (cm)	37.0 ± 2.5	37.5 ± 3.6	39.9 ± 3.4	<0.01 ^b^
Mean SpO_2_ (%)	96.6 ± 1.1	95.9 ± 1.1	91.5 ± 4.4	<0.01 ^b^
Lowest SpO_2_ (%)	89.3 ± 5.8	84.6 ± 4.5	75.3 ± 10.2	<0.01 ^b^
Oxygen desaturation index (≥3%, events/h)	3.3 ± 3.9	8.8 ± 7.9	44.8 ± 23.1	<0.01 ^b^
AHI (events/h)	8.4 ± 3.7	21.5 ± 4.5	54.0 ± 18.5	<0.01 ^b^

OSA: obstructive sleep apnoea; PSG, polysomnography; SpO_2_: pulse oxygen saturation; AHI: apnoea–hypopnea index; ns: nonsignificant. Data are expressed as means ± standard deviations. ^a^ One-way analysis of variance; ^b^ Kruskal–Wallis test; ^c^ Chi-squared test.

**Table 2 sensors-21-08097-t002:** Comparison of the sleep parameters obtained by RootiRx in hospital and home settings.

Variables	Group	Hospital	Home	*p* Value
CVHRI (events/h)	No-to-mild, *n* = 33	10.3 ± 8.1	9.3 ± 9.9	0.50 ^a^
	Moderate, *n* = 31	11.8 ± 8.5	10.1 ± 7.0	0.41 ^b^
	Severe, *n* = 61	33.8 ± 21.1	20.4 ± 18.2	<0.01 ^a^
CEI (events/h)	No-to-mild, *n* = 33	5.4 ± 4.4	5.0 ± 2.7	0.78 ^a^
	Moderate, *n* = 31	9.2 ± 4.4	7.6 ± 4.1	0.16 ^a^
	Severe, *n* = 61	18.2 ± 12.2	13.1 ± 9.9	<0.01 ^a^
Rx index (events/h)	No-to-mild, *n* = 33	14.2 ± 8.6	14.0 ± 8.6	0.79 ^a^
	Moderate, *n* = 31	18.4 ± 8.0	16.0 ± 6.6	0.20 ^a^
	Severe, *n* = 61	40.9 ± 21.1	27.1 ± 18.4	<0.01 ^a^
Supine sleep time (%)	No-to-mild, *n* = 33	72.5 ± 27.0	58.0 ± 17.9	<0.01 ^a^
	Moderate, *n* = 31	77.7 ± 21.4	56.0 ± 18.0	<0.01 ^a^
	Severe, *n* = 61	74.1 ± 23.9	48.9 ± 21.9	<0.01 ^a^

PSG: polysomnography; CVHRI: cyclic variation of heart rate index; CEI: chest effort index; Rx index: combination of the CVHRI and CEI; ns: nonsignificant. Data are expressed as means ± standard deviations. All *p* values were derived from the ^a^ Mann–Whitney U test or ^b^ Student’s *t*-test depending on whether the data sets met the normality assumptions.

**Table 3 sensors-21-08097-t003:** Comparison of the home- and hospital-based RootiRx results in the high-sleep-quality groups (sleep efficiency ≥80%).

Variables	Group	Hospital	Home	*p* Value
CVHRI (events/h)	No-to-mild, *n* = 24	10.14 ± 8.66	9.36 ± 10.36	0.70 ^a^
	Moderate, *n* = 24	12.15 ± 7.34	9.88 ± 6.65	0.27 ^b^
	Severe, *n* = 55	34.39 ± 21.27	20.17 ± 17.89	< 0.01 ^a^
CEI (events/h)	No-to-mild, *n* = 24	6.02 ± 4.91	5.10 ± 2.98	0.90 ^a^
	Moderate, *n* = 24	9.08 ± 4.65	7.00 ± 3.12	0.17 ^a^
	Severe, *n* = 55	18.22 ± 11.44	13.58 ± 10.21	< 0.05 ^a^
Rx index (events/h)	No-to-mild, *n* = 24	14.45 ± 9.28	14.06 ± 9.29	0.83 ^a^
	Moderate, *n* = 24	18.55 ± 7.22	15.48 ± 6.00	0.12 ^b^
	Severe, *n* = 55	41.42 ± 21.01	27.16 ± 18.41	< 0.01 ^a^
Supine sleep time (%)	No-to-mild, *n* = 24	69.40 ± 28.43	54.93 ± 19.26	0.02 ^a^
	Moderate, *n* = 24	80.51 ± 19.85	57.60 ± 17.91	< 0.01 ^a^
	Severe, *n* = 55	75.07 ± 23.81	48.97 ± 22.28	< 0.01 ^a^

CVHRI: cyclic variation of heart rate index; CEI: chest effort index; Rx index, combination of CVHRI and the CEI; ns: nonsignificant. Data are expressed as means ± standard deviations. All *p* values were derived from the ^a^ Mann–Whitney U test or ^b^ Student’s *t*-test depending on whether the data sets met the normality assumptions.

## Data Availability

The datasets used and analysed during the current study are available from the corresponding author on reasonable request.

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
