# Peer review of "Comparison of Hospital-Based and Home-Based Obstructive Sleep Apnoea Severity Measurements with a Single-Lead Electrocardiogram Patch"

_sensors, 2021, doi:10.3390/s21238097_

Round 1
Reviewer 1 Report
The revised manuscript is more convincing than the first version from a statistical standpoint. Nonetheless, the paper needs more clarity in its statements and claims. For instance, the final paragraph in the introduction reads, "The primary objective of this study was to compare the data on sleep parameters obtained in overnight PSG at the hospital and over several days at home through a single-lead ECG patch with a 3-axis accelerometer (RootiRx), with the results expected to provide an in-depth understanding of how sleep position and the environment affect OSA severity". The obvious question is, what are these reference results that provide a higher understanding of sleep position influence. Moreover, in the conclusions, the authors interpret their results very euphemistically, overusing terms like "may" or "possibly" (e.g., "The current results indicate evidence to support that patients may spend a higher percentage of sleep time in the supine position in hospital settings, which may possibly result in the OSA severity overestimation").
Indeed, the approached subject is complex, and it is hard to draw clear-cut conclusions. Given the circumstances, avoiding being vague and fuzzy can be achieved by:
- Formulating an enumeration of this paper's contributions to the field, crisp and clear, at the end of the introduction.
- Adding a "Limitations" subsection to the discussions, where the authors clearly describe the main constraints and pitfalls facing their study. This way, one may understand the need to refrain from formulating strong/solid claims.
Reviewer 2 Report
Paper is about to compare hospital- and home-based sleep parameters by utilizing the 382 RootiRx, a wearable device that uses a single-lead ECG patch. Followings are my findings:
- Intro section lacks about methodology and contribution. Just a purpose of study is given.
- Too much abbreviations bores the reader outside from medicine.
- Very detailed related works section is expected.
- Table 2 is problematic about the format. Kind of line numbers are added.
- Figure 3 and caption should be corrected.
- Explenations about the figures and tables are limited. Giving more detail would be appreciated.
- There are several typos, Language should be improved.
Round 2
Reviewer 2 Report
All my reviewer points are adressed.
This manuscript is a resubmission of an earlier submission. The following is a list of the peer review reports and author responses from that submission.
Round 1
Reviewer 1 Report
This paper addresses an essential subject for sleep and respiratory medicine, namely the reliable OSA diagnosis. The subject is relevant for the Sensors journal, as sleep/respiratory medicine investigations rely on sensor-based physiological signal time series recordings. In this context, the authors compare the hospital- and home-based polysomnography in correlation with the supine sleep position. The paper is well written, and the findings are presented with clarity.
However, there are two critical concerns that the authors need to address.
- The main contribution of the paper is the finding that the supine position influences the sensor measurements. In particular, the hospital environment fosters this sleep position and, therefore, alters the diagnostic outcome, resulting in an overestimation of the OSA stage. However, the subject of positional OSA is well known in the literature and that the supine position induces OSA https://jcsm.aasm.org/doi/10.5664/JCSM.1194. Moreover, the literature also approaches the difference the diagnosis environment (i.e., hospital vs. home) makes on sleep position during polysomnography https://link.springer.com/article/10.1007/s11325-020-02099-w. Consequently, the authors need to put their contribution into context thoroughly. Otherwise, the impact of their findings cannot be assessed.
- The authors present various correlations between the supine position sleep percentage and various OSA indexes. Just by eyeballing Figures 3 and 5, one may wonder what the r values will be if we consider a few measurement errors. The statistical solution to this concern is to calculate and present the confidence intervals of the correlations.
- The authors use more than one device type for their measurements. Are all patients measured with all these devices? Is there any inter and intra-variability of the measurements concerning the device types?
In addition, the authors also need to undertake the following actions to improve the presentation of their paper.
- The authors use medical terms such as AHI, SE, etc., but do not explain them, although it is difficult to define them clearly. For instance, AHI is the number of apneas/hypopneas occurring per hour of sleep. Please clearly/carefully explain these medical terms because Sensors is not a respiratory/sleep medical journal.
- The authors say, "Moreover, this finding has clinical relevance because if sleeping position is influenced by the PSG apparatus and this causes significant overestimation of OSA severity, treatment strategies are likely to be affected." As the standard treatment for OSA is CPAP, the question is how this overestimation make any difference for the CPAP.
Reviewer 2 Report
This is an interesting work that follows up on a previous work published by the same authors which introduced a single-lead ECG patch for at-home diagnosis of sleep apnea. The ability to evaluate sleep parameters at home using a minimally obstructive device can indeed help alleviate the discomfort often reported in polysomnography studies.
The authors reported a significant positive correlations between variations in "supine percentage" and OSA indices and important conclusions were drawn based on these correlation results. However, the values of the spearman rank were relatively low and indicate rather no correlation (significance is independent from the strength on correlation analysis) between the compared parameters, which would eventually lead to different result interpretations and possibly different conclusions. The authors should clarify why and how they considered these values to indicate strong correlations, and potentially consider revising the conclusions.